# The Emerging Roles of the Adaptive Immune Response in Acute Pancreatitis

**DOI:** 10.3390/cells12111495

**Published:** 2023-05-29

**Authors:** Bojan Stojanovic, Ivan P. Jovanovic, Milica Dimitrijevic Stojanovic, Marina Jovanovic, Berislav Vekic, Bojan Milosevic, Aleksandar Cvetkovic, Marko Spasic, Bojana S. Stojanovic

**Affiliations:** 1Department of Surgery, Faculty of Medical Sciences, University of Kragujevac, 34000 Kragujevac, Serbia; bojan.stojanovic01@gmail.com (B.S.);; 2Center for Molecular Medicine and Stem Cell Research, Faculty of Medical Sciences, University of Kragujevac, 34000 Kragujevac, Serbia; 3Department of Pathology, Faculty of Medical Sciences, University of Kragujevac, 34000 Kragujevac, Serbia; 4Department of Internal Medicine, Faculty of Medical Sciences, University of Kragujevac, 34000 Kragujevac, Serbia; 5Department of Pathophysiology, Faculty of Medical Sciences, University of Kragujevac, 34000 Kragujevac, Serbia

**Keywords:** acute pancreatitis, immune response, T cells, B cells

## Abstract

Acute pancreatitis (AP) is an abrupt, variable inflammatory condition of the pancreas, potentially escalating to severe systemic inflammation, rampant pancreatic necrosis, and multi-organ failure. Its complex pathogenesis involves an intricate immune response, with different T cell subsets (Th1, Th2, Th9, Th17, Th22, TFH, Treg, and CD8^+^ T cells) and B cells playing pivotal roles. Early T cell activation initiates the AP development, triggering cytokines associated with the Th1 response, which stimulate macrophages and neutrophils. Other T cell phenotypes contribute to AP’s pathogenesis, and the balance between pro-inflammatory and anti-inflammatory cytokines influences its progression. Regulatory T and B cells are crucial for moderating the inflammatory response and promoting immune tolerance. B cells further contribute through antibody production, antigen presentation, and cytokine secretion. Understanding these immune cells’ roles in AP could aid in developing new immunotherapies to enhance patient outcomes. However, further research is required to define these cells’ precise roles in AP and their potential as therapeutic targets.

## 1. Introduction

Acute pancreatitis (AP) is an inflammatory condition that occurs after premature activation of pancreatic proteolytic enzymes [1,2]. This condition can be stratified into the following two primary stages: interstitial edematous pancreatitis and necrotizing pancreatitis [1]. Interstitial edematous pancreatitis, the milder form, involves inflammation and edema of the pancreas, but no tissue death [3]. In contrast, necrotizing pancreatitis involves inflammation and widespread death of pancreatic tissue, and it can further progress into severe pancreatitis if complications occur [4]. Severe pancreatitis typically presents with persistent organ failure, which is defined by a modified Marshall scoring system. This can manifest as acute kidney injury, respiratory distress syndrome, or even multiple organ dysfunction syndrome. Without early diagnosis and appropriate intervention, the mortality rate in severe pancreatitis can be significantly high [1,5].

From the very beginning, AP is accompanied by numerous immune system alterations [1,2]. Compelling evidence has emerged for the role of multiple types of innate immune cells, including neutrophils, monocytes/macrophages, mast cells, and dendritic cells, in the development of AP and the occurrence of local and systemic complications [6]. Besides innate immune cells, abnormal activation of adaptive immune cells is a significant event in immune response modulation and AP progression [7]. Namely, a hyperinflammatory process develops after injury to pancreatic acinar cells, infiltrating innate immune cells, particularly neutrophils, and macrophages. This process can lead to the development of a systemic inflammatory response (SIRS) and potentially even multi-organ dysfunction syndrome (MODS) [8]. The subsequent phase involves the emergence of a compensatory anti-inflammatory response (CARS), in which T cells play a crucial role in protecting against hyperinflammation [9].

Lymphocytes are mononuclear white blood cells that participate in numerous functions of the immune system, such as defense, immune surveillance, and immune regulatory function by which they suppress the excessive production of pro-inflammatory mediators [10]. Adaptive immune cells can be categorized into the following two subsets according to their function and the markers present on their surface: T cells (T lymphocytes) and B cells (B lymphocytes) [11]. Excessive activation of adaptive immune cells during the early stages of AP may contribute to the development of MODS [12].

During AP progression, changes in the number of lymphocytes occur at various stages [7]. At the onset of AP, there is an increase in the expression of markers CD25, CD38, CD28, and CD122 on the surface of cells of acquired immunity, indicating an activated phenotype of T and B cells [12,13]. As AP progresses, there is a significant increase in the number of leukocytes, while the total count of T cell and B cell count decreases when compared to healthy individuals [14,15]. In the later stages of AP, the number of circulating lymphocytes recovers, except in complicated forms of the disease with the development of infection [13].

Comparisons between mild acute pancreatitis (MAP) and severe acute pancreatitis (SAP) reveal differences in lymphocyte activation. Sweeney et al. [16] observed abnormal T cell activation in MAP but not in SAP, a more severe form of the disease associated with systemic and local complications. Furthermore, the reduction in the total number of T and B cells is more pronounced in SAP than in MAP. This reduction is particularly persistent for B cells, which remain decreased even one month after the onset of AP [17,18]. These findings underscore the unique changes to cellular and humoral immunity in different forms and stages of AP, and their potential implications for disease progression and recovery.

This review aims to provide an account of the diverse populations of adaptive immune cells and their respective roles in the pathogenesis of AP.

## 2. T Cells

The cells of adaptive immunity, including T cells, are among the main effector cells that contribute to the development of the inflammatory response during AP [19]. T cells comprise several subsets with different functions, such as T helper cells (Th cells), regulatory T cells (Tregs), and cytotoxic T cells [20]. Upon the onset of pancreatitis, the pancreas experiences the significant infiltration of neutrophils and monocytes/macrophages, together forming the predominant leukocyte population in the affected organ. Concurrently, an imbalance of T lymphocytes, encompassing both CD4^+^ and CD8^+^ T cells, has been detected in the inflamed pancreas as well as in the bloodstream of patients suffering from pancreatitis [6,21,22].

During the early stages of AP, there is a decrease in the number of T cells in circulation, which includes both CD4^+^ and CD8^+^ T cells, regardless of the disease severity [17,23,24]. This decline in T cell numbers is caused by the heightened apoptosis of circulating T lymphocytes, which occurs through the signaling pathway of Fas/FasL in T cells (Figure 1) [13]. Clinical research indicates that excessive expression of Fas/Fas ligands can significantly reduce circulating CD4^+^ T cells and cause a decline in the CD4^+^/CD8^+^ ratio [24]. This decrease in the CD4^+^/CD8^+^ ratio indicates a more significant loss of CD4^+^ T cells in comparison to CD8^+^ T cells. This finding suggests that a quantitative impairment of T lymphocytes, including both CD4^+^ and CD8^+^ T cells, could be closely associated with infectious complications observed during SAP. Moreover, abnormal expression of Fas in peripheral blood is linked to excessive apoptosis of T lymphocytes and a significant loss of T lymphocyte subsets, ultimately leading to immunosuppression and the onset of sepsis [24].

Experimental studies have demonstrated a direct link between the overexpression of FAS signaling and the severity of AP, as well as the development of infectious complications. This occurs through the promotion of T cell apoptosis, affecting both CD4^+^ and CD8^+^ T cells [25,26]. To clarify, the reduction in total T cells, including both CD4^+^ and CD8^+^ T cells, occurs early in AP, irrespective of the severity [17,23,24]. However, a more pronounced reduction in T cells, especially CD4^+^ T cells, is observed in severe cases such as SAP [24]. The studies mentioned focus on the overall reduction in circulating T cells in the context of AP severity and complications, and further research is needed to explore the potential differences between specific T cell types and organ-specific locations in AP.

In addition to increased apoptosis, reduced numbers of circulating T cells during AP may also be due to the migration of activated lymphocytes to inflamed areas such as the pancreas and lungs, as well as lymphocyte homing (Figure 1) [9]. The process of T cell homing to inflamed areas is mediated by a series of adhesion molecules and chemokines that guide T cells to the site of inflammation [27]. T cell egress from lymph nodes to inflamed tissue involves the following three steps: selectin-dependent rolling, chemokine-mediated activation, and integrin-dependent arrest [28]. Therefore, despite the overall depletion of circulating T cells, T cells guided to the pancreas by these signals can still reach the inflamed tissue.

Although the precise mediators responsible for T cell homing to the inflamed pancreas are not fully understood, selectins, integrins, and chemokines are known to play crucial roles in T cell homing to other inflamed tissues [27]. Various key molecules, such as P-selectin, E-selectin, P-selectin glycoprotein ligand-1 (PSGL-1), α4β1 integrin (very late antigen-4 or VLA-4), αLβ2 integrin (lymphocyte function-associated antigen-1 or LFA-1), vascular cell adhesion molecule-1 (VCAM-1), intercellular adhesion molecule-1 (ICAM-1), and chemokine receptors such as CXCR3 and its ligands, CXCL9, CXCL10, and CXCL11, are also believed to be involved in T cell homing to the pancreas during acute pancreatitis [29]. Further research is needed to identify the specific adhesion molecules and chemokines that govern T cell homing to the inflamed pancreas and their underlying mechanisms.

The experimental study revealed that CD4^+^ and CD8^+^ T cells infiltrate the inflamed pancreas, with CD4^+^ T cells being more prevalent in the adjacent acini [22]. The local microenvironment in the inflamed pancreas may offer some protection against apoptosis or slow down the rate of apoptosis compared to circulation. Additionally, T cells at different activation or differentiation stages could have varying susceptibilities to apoptosis, potentially contributing to differences between T cells in circulation and those in the pancreas. However, further research is necessary to fully elucidate the mechanisms underlying T cell viability and function in both circulation and inflamed tissues.

### 2.1. CD4^+^ T Cells

The number of circulating CD4^+^ T helper (Th) cells is typically lower in patients with AP upon admission, but it increases significantly within five days from disease onset [30]. However, a decreased number of CD4^+^ T cells can predict the development of severe AP with organ failure in the later stages of the disease, with a sensitivity of 61.54% and specificity of 90% [30,31]. In patients with more severe disease courses, such as those with local complications such as pseudocysts, sterile necrosis, or infected necrosis, the percentage of Th cells remains significantly lower [21]. While the increase in CD4^+^ T cell numbers within five days of onset may be due to the natural progression of less severe cases, patients with severe AP and organ failure may not exhibit the same recovery in CD4^+^ T cell numbers. Further research is needed to fully understand the dynamics of CD4^+^ T cell numbers during AP and their association with various treatments and patient outcomes.

Additionally, the decrease in the number of circulating CD4^+^ T cells in the initial phase of acute pancreatitis is significantly associated with the development of abdominal compartment syndrome (ACS) [32]. ACS is a life-threatening condition characterized by increased intra-abdominal pressure, leading to reduced blood flow to abdominal organs, organ dysfunction, and ultimately, multi-organ failure [33]. In the context of AP, ACS is relevant because it can worsen disease severity, cause complications, and negatively impact patient outcomes [34]. The link between decreased CD4^+^ T cell numbers and the development of ACS emphasizes the potential importance of immune system alterations in the progression and severity of AP.

Moreover, experimental studies have shown a reduction in the number of Th cells in the intestinal mucosal lamina propria during the initial days of severe acute pancreatitis induction, which leads to a significant decline in intestinal immune functions. As a consequence, there is an increased translocation of bacteria and endotoxin after the initiation of experimental SAP [35].

The pattern of cytokine secretion and activation of transcription factors determines the differentiation of naïve T cells into various Th cell phenotypes in a particular environment. T helper cell phenotypes include Th1, Th2, Th9, Th17, and Th22 cells, along with follicular helper T cells (TFH) and regulatory T (Treg) cells [36]. In addition, regulatory T cells have the following two subsets based on their origin: inducible Treg cells and natural Treg cells [37].

#### 2.1.1. Th1 and Th2 Cells

Th1 cells are known to predominantly synthesize IFN-γ, which stimulates the effector functions of macrophages [38]. In contrast, Th2 cells synthesize mediators such as IL-4, IL-5, and IL-13, which modulate the functions of multiple immune cells, including eosinophils, basophils, mast cells, and B cells, and enable parasite resistance to infection [39]. During the initial phase of SAP, there is a significant decrease in the number of circulating Th1 cells, while the number of circulating Th2 cells increases in the first few days [40]. The Th1/Th2 ratio is also reported to decrease during the first week of AP, with the suppression of Th1 cells and increased Th2 cell activity [40]. The initial suppression of Th1 cells and the predominance of Th2 responses, leading to an imbalance in Th1/Th2 cells, may be related to the induction of SAP’s systemic inflammatory response characteristic [9]. However, with AP progression, Th1 response is induced and the Th1/Th2 ratio increases [9]. In addition, it has been well-propounded that reduced Th1 cells are markedly activated in SAP, and markedly higher concentrations of IFN-γ, IL-6, and TNF-α are released during SAP [41]. In contrast, increased concentrations of IL-4 and IL-13 have been observed in mild/moderately severe forms of AP, indicating significant involvement of Th2 cells in MAP development [41]. In this context, the dynamic imbalance between Th1/Th2 cells contributes to the occurrence of immunopathogenic events associated with SAP development, with a shift from Th1 to Th2 cells observed during the course of the disease [9]. Therefore, maintaining the balance of Th1 and Th2 responses might be a promising therapeutic strategy for preventing AP progression. Furthermore, studies have shown a predominance of cells with an anti-inflammatory phenotype in lymphoid tissue surrounding the pancreas, mainly due to the abundance of Th2 cells and Tregs [42].

#### 2.1.2. Th9 Cells

Th9 cells have recently been identified as a subtype of CD4^+^ cells that display both pro-inflammatory and Th2 cell-like features [43]. These cells produce IL-9 as their signature cytokine, which facilitates the recruitment of eosinophils, basophils, and mast cells, leading to the development of autoimmune diseases, tumors, and infections [44]. During AP, an increased influx of eosinophils, basophils, mast cells, and other innate immune cells into the pancreatic parenchyma has been reported [6,45]. The observed data indicate that Th9 cells might play a role in shaping the immune responses that contribute to AP development. However, no studies to date have monitored the levels of circulating Th9 cells or their infiltration into the pancreas in the context of AP. Furthermore, Merilainen et al. [46] found no significant difference in serum IL-9 concentrations between experimental edematous and necrotizing pancreatitis groups and sham controls during the early course of AP. Therefore, further research is needed to clarify the potential role of Th9 cells in the pathogenesis of AP and their potential as a therapeutic target.

#### 2.1.3. Th17 Cells

Th17 cells are characterized by their production of IL-17, a pro-inflammatory cytokine that is also produced by other cell types, such as NK cells, innate lymphoid cells type 3, and NKT cells [47]. IL-17 induces the production of granulocyte colony-stimulating factor and IL-8 upon binding to receptors on target cells, including innate immune cells and epithelial cells, which in turn promote the influx of activated neutrophils into inflamed tissues [48]. Neutrophils are the first cells to infiltrate the pancreas following acinar cell damage, producing oxidants and cytotoxic mediators that contribute to local pancreatic tissue damage and distant organ system damage, such as acute respiratory distress syndrome (ARDS) in the lungs [6]. Th17 cells play a crucial pro-inflammatory role in AP development and serve as a significant prognostic marker for assessing disease severity in AP patients [49]. This is supported by findings showing increased serum IL-17 concentrations that correlate with AP severity [49]. The Th17 response, known for its pro-inflammatory nature, has been shown to play a significant role in initiating early SIRS in AP. This is accomplished through mechanisms such as the enhanced inflammatory cascade, increased neutrophil infiltration, macrophage recruitment to inflammatory sites, and elevated production of inflammatory molecules and cytokines, leading to pancreatic injury [43,50]. Recent human studies have also revealed a significant correlation between the percentage of circulating IL-17^+^ cells and SAP severity [51]. Therefore, the number and activation degree of circulating Th17 cells can serve as prognostic markers for SAP patients [52,53]. Targeting IL-17, IL-23, or IL-6 has been shown to reduce pancreatic and systemic inflammation in experimental AP models by the therapeutic suppression of the Th17 response [53]. Further studies are needed to evaluate the efficacy and safety of these approaches in clinical settings.

#### 2.1.4. Th22 Cells

There is a subset of effector CD4^+^ T cells called Th22 cells that are characterized by the production of signature cytokine IL-22, as well as IL-23 and TNF-α [54]. While the pro-inflammatory and anti-inflammatory effects of Th22 cells are still under investigation, they are believed to play a role in the pathogenesis of various diseases, including inflammatory diseases, autoimmune disorders, and tumors [55]. Elevated plasma concentrations of IL-22 have been reported in AP patients, and the administration of recombinant IL-22 has been found to reduce the severity of experimental AP [56,57]. However, in contrast to these findings, recent animal model studies have found lower expression levels of IL-22 mRNA and a reduced presence of Th22 cells in the lungs following the induction of experimental AP [58]. The discrepancy between these findings can be attributed to the complex and context-dependent roles of Th22 cells and IL-22 in different tissues and disease stages. Elevated plasma levels of IL-22 during AP may reflect a systemic response to inflammation, while the reduced expression of IL-22 in the lungs could be due to localized tissue-specific factors or the suppression of Th22 response in certain conditions. Furthermore, other immune cells, such as NK cells, innate lymphoid cells, and NKT cells, can also produce IL-22, potentially contributing to the increased plasma levels of IL-22 in AP patients [59].

The protective role of Th22 cells in AP development has been proposed to involve the modulation of autophagy pathways [57]. Specifically, Th22 cells have been suggested to induce the expression of anti-apoptotic proteins Bcl-2 and Bcl-XL via the production of IL-22. These proteins bind to Beclin-1, a key autophagy regulator, inhibiting the formation of autophagosomes and preventing the initiation of autophagy in pancreatic cells. This mechanism may help to reduce pancreatic cell damage and inflammation, potentially contributing to a protective effect of Th22 cells and IL-22 in the context of AP [57,60]. Additionally, recent studies have shown the pivotal role of the transcription factor aryl hydrocarbon receptor (AhR) in promoting pancreatic Th22 response [61]. AhR is a ligand-activated transcription factor that promotes the differentiation of naïve CD4^+^ T cells into Th22 cells upon activation, thus enhancing the Th22-mediated response in the context of AP [61,62]. Nevertheless, further research is necessary to fully understand the roles of Th22 cells and IL-22 in AP and to explore their potential as therapeutic targets.

#### 2.1.5. TFH Cells

T follicular helper (TFH) cells play a crucial role in promoting and maintaining germinal centers and are primarily localized in secondary lymphoid organs [63]. TFH cells express high levels of the programmed cell death protein 1 (PD-1) and play a significant role in promoting humoral immune responses [64]. This includes supporting class-switching in germinal centers and facilitating the maturation and differentiation of B lymphocytes [65]. TFH cells have been reported to play an important role in inflammation, autoimmunity, and infection [66]. The key cytokine secreted by TFH cells is interleukin-21 (IL-21) [65].

Increased mRNA expression and plasma concentrations of IL-21 have been observed in patients with SAP compared to those with MAP [67]. IL-21 is believed to contribute to the progression of pancreatitis by modulating immune responses and promoting inflammation [68]. In the late disease stages, transiently increased levels of IL-21 have been reported in patients who develop septic complications and pancreatic necrosis, which contributes to immune paresis due to the inhibition of regulatory T cells (Tregs) [67,69]. Therefore, TFH cells may play a significant role in the progression of acute pancreatitis. However, additional studies are needed to elucidate the precise role of TFH cells and the impact of IL-21 in the context of acute pancreatitis, particularly in terms of the potential therapeutic use of IL-21 to regulate immune responses in AP.

#### 2.1.6. Treg Cells

The primary characteristic of Treg cells implies regulating and suppressing immune responses, particularly those that could lead to excessive inflammation or autoimmunity [70]. Treg cells achieve this by reducing the activity of other immune cells, including dendritic cells and NK cells, and promoting macrophage differentiation into M2 phenotypes with anti-inflammatory characteristics [71]. Therefore, Tregs play a role in controlling inflammatory damage after a severe injury in SAP [72]. The number of Treg cells in AP is controversial. Recent research has suggested that the number of Tregs continuously increases during AP progression, which could be related to immune suppression, a characteristic of CARS [73]. Moreover, a sustained increase in the proportion of circulating Treg cells has been associated with secondary immune activation, as evidenced by the elevated secretion of anti-inflammatory cytokines such as IL-10 and TGF-β [74,75].

Minkov et al. [76] showed that a high percentage of natural Treg cells is accompanied by an unfavorable response. Furthermore, previous reports indicate different biomarkers of Treg cells, such as CD4^+^CD25^+^CD127^−/neg^ or CD4^+^CD25^+^Foxp3^+^, and their association with different phenotypes of Tregs and forms of AP. Thus, an increased number of circulating CD4^+^CD25^+^CD127^low/neg^ Tregs is associated with an increased risk of developing infectious complications in SAP and mortality [76]. Conversely, the level of circulating CD4^+^CD25^+^CD127^high^ Tregs is lower in patients who develop multiple organ failure in the early phase of AP [77]. In line with previous reports, the induction of experimental AP was associated with a reduced number of CD4^+^CD25^+^ T cells in comparison with sham controls. The reduced number of Tregs promoted the activity of effector CD4^+^ T cells that infiltrate the pancreas, which promotes pancreatic necrosis [22]. We found that the influx of CD4^+^CD25^+^Foxp3^+^ T cells is lower in pancreatic tissue after the induction of experimental AP, which probably occurs as a consequence of the reduced expression of tolerogenic dendritic cells (data not published) due to the dominant pro-inflammatory milieu that develops in the first days following AP initiation [78]. In a previous study, Zheng et al. [75] reported that nicotine use markedly decreases the mortality of SAP patients, which is related to an increase in the number and suppressive capacity of CD4^+^CD25^+^ T regulatory cells. Experimental studies have demonstrated that in acute pancreatitis, activation of Tregs can disrupt duodenal barrier function, leading to the translocation of commensal bacteria into pancreatic necrosis [79]. To date, there is very little data on Treg cells in human studies, and it is necessary to determine the precise role of Tregs in the progression of AP.

### 2.2. CD8^+^ T Cells

CD8^+^ T cells, also known as cytotoxic T lymphocytes, are effector cells of acquired immunity that play a crucial role in eliminating intracellular bacteria, viruses, and cancer cells through cell–cell interactions or the secretion of perforin and granzyme [80]. Alterations in CD8^+^ T cell numbers during acute pancreatitis remain unclear due to conflicting observations in the development of MAP or SAP [7]. An increase in soluble CD8 levels has been reported in AP cases [81]. However, some studies present contrasting findings; Dabrowski et al. [82] demonstrated a decrease in the absolute number of cytotoxic T lymphocytes in peripheral blood, whereas Pinhu et al. [25] observed an increased number of cytotoxic T lymphocytes. These discrepancies in CD8^+^ T cell dynamics during AP highlight the complexity of the immune response and warrant further investigation to better understand the role of CD8^+^ T cells in AP progression.

Among the CD8^+^ T cell populations, there are phenotypes with immunosuppressive characteristics that inhibit antibody secretion or cellular immunity [83]. Additionally, CD8^+^ T regulatory cells secrete anti-inflammatory cytokines, such as IL-10 and TGF-β [84]. These immunosuppressive phenotypes of CD8^+^ T cells are involved in numerous diseases, such as autoimmune diseases or tumors. However, no reports have specifically investigated the connection between these cells and the development of AP. Further studies are needed to explore the potential roles of CD8^+^ T cell subsets, including immunosuppressive phenotypes, in the pathogenesis and progression of AP.

## 3. B Cells

The primary function of B cells is to secrete antibodies, which are essential for humoral immunity in acquired immunity [10]. In patients with AP and diagnosed OF, the number of B lymphocytes is significantly elevated [30]. Additionally, the number of CD19^+^ B cells in the serum of AP patients has been found to have predictive value for the development of OF, with greater numbers of activated B cells indicating a more severe systemic inflammatory response and a higher likelihood of OF occurrence [30]. Conversely, a reduced number of CD19^+^ B cells has been observed in both the early and late phases of severe AP, with a more pronounced reduction associated with the development of infectious complications [17]. The length of hospital stay in patients with mild AP has also been positively related to the number of B cells [14].

Subsequently, the production of immunoglobulin—the main effector molecule produced by B cells—is altered in AP patients compared to healthy controls. Specifically, serum levels of IgM and IgG are significantly reduced in patients with infectious complications, and IgG levels are markedly lower in patients with fatal outcomes. In contrast, serum levels of IgA remain unchanged [15,85]. These findings suggest that in severe forms of AP, B cell function is suppressed, which can lead to the chronic impairment of humoral immunity.

Furthermore, the induction of experimental AP in B-cell-deficient mice was accompanied by a significant increase in the histological score of AP, more pronounced pancreatic edema, and markedly increased serum amylase values [86]. B cells are believed to play a significant immunomodulatory role in the development of AP and inhibit the activation and proliferation of other immune cells by secreting anti-inflammatory mediators [87]. Consistently, this immunosuppressive function is attributed to the activity of regulatory B cells that produce IL-10, IL-35, and TGF-β and prevent excessive activation and expansion of pro-inflammatory cells [88]. There are two subtypes of regulatory B cells, which are as follows: memory CD19^+^CD24^hi^CD27^hi^ B cells and immature/transitional CD19^+^CD24^hi^CD38^hi^ B cells [89,90,91]. Importantly, CD19^+^CD24^hi^CD38^hi^ cells inhibit Th1 and Th-17 pro-inflammatory immune responses by producing IL-10 [89].

Furthermore, a depletion of circulating CD19^+^CD24^hi^CD27^hi^ cells that produce IL-10 has been observed in AP patients, and this depletion is more pronounced in patients with a severe form of the disease [92]. On the other hand, in MAP patients, the number of both subsets of regulatory B cells was significantly increased from the first to the seventh day of the disease, which indicates the protective role of these subsets of B cells in reducing disease severity, while the decrease in the number can play a role as a SAP predictor [92].

## 4. Differences in T Cell and B Cell Responses between Mild Acute Pancreatitis (MAP) and Severe Acute Pancreatitis (SAP)

Understanding differences in T cell and B cell responses between MAP and SAP is essential in identifying potential therapeutic targets and designing personalized treatment strategies. A comprehensive comparison of immune responses in these two clinical presentations can offer valuable insights into the factors contributing to AP severity and progression.

### 4.1. T Cell Responses in MAP vs. SAP

Several studies have reported differences in T cell subsets between MAP and SAP patients. For instance, the number of circulating CD4^+^ T cells is generally lower in SAP patients compared to those with MAP, and the activation status of these cells can also differ between the two groups [30,31]. Furthermore, Th17 cells have been found to play a more significant pro-inflammatory role in SAP, with higher serum concentrations of IL-17 correlating with increased disease severity [49]. In contrast, Th22 cells are believed to play a protective role in AP, with their activation and expression differing between MAP and SAP patients [57,58]. Finally, the role of Treg cells and their immunosuppressive function may vary between MAP and SAP, with more pronounced dysfunction observed in patients with severe disease [74,77].

### 4.2. B Cell Responses in MAP vs. SAP

Although the role of B cells in AP is less well studied, some differences between MAP and SAP have been observed. For example, in patients with MAP, the number of B cells has been positively correlated with the length of hospital stay [14], whereas in SAP, a suppression of B cell function is observed, leading to the chronic impairment of humoral immunity [15,85]. Furthermore, SAP patients have been reported to have higher levels of B cell-activating factor (BAFF) in their plasma, which can contribute to B cell survival, proliferation, and differentiation [93]. Specifically, regulatory B cells, which are crucial for immunomodulation, show contrasting patterns in MAP and SAP [89,90,91,92]. In MAP, the number of both subtypes of regulatory B cells significantly increases from the first to the seventh day of the disease, indicating a protective role in reducing disease severity [92]. Conversely, in SAP, there is a marked depletion of circulating regulatory B cells that produce IL-10, an anti-inflammatory cytokine [92].

There are notable differences in the immune responses of T cells and B cells between MAP and SAP patients, as illustrated in Figure 2 and Figure 3, which depict the role of T and B cell responses in the development of severe and mild acute pancreatitis. Further investigation of these differences can help elucidate the underlying mechanisms that contribute to the severity and progression of AP, ultimately informing the development of targeted therapies and personalized treatment approaches.

Figure 2 and Figure 3 highlight the key differences in T cell and B cell responses between MAP and SAP, focusing on the balance between pro-inflammatory and anti-inflammatory immune responses. In MAP, the immune response is predominantly anti-inflammatory, with regulatory T and B cells maintaining immune tolerance. In contrast, SAP is characterized by a primarily pro-inflammatory immune response, with a relative deficiency of regulatory T and B cells leading to uncontrolled inflammation and tissue damage. Understanding these differences in immune responses may aid in the development of targeted therapies to modulate the immune response and improve outcomes for patients with AP.

## 5. Potential Therapeutic Strategies and Future Directions

The complexity and heterogeneity of the immune response during acute pancreatitis present numerous challenges in developing effective treatments. Addressing the major gaps in understanding and expanding upon specific immunotherapies are crucial steps in propelling the field forward. This section discusses potential therapeutic strategies and areas for future research, particularly focusing on the distinction between mild acute pancreatitis and severe acute pancreatitis.

### 5.1. Targeted Immunotherapies

Targeted immunotherapies are emerging as potential treatment options for AP. These therapies aim to modulate the immune response during AP to alleviate disease severity and prevent complications. For instance, the suppression of Th17 cells by targeting IL-17, IL-23, or IL-6 has shown promise in reducing pancreatic and systemic inflammation in experimental models of AP [53]. Similarly, enhancing the protective role of Th22 cells and IL-22 may offer another therapeutic avenue, particularly through the modulation of autophagy pathways [56,57]. However, further studies are needed to evaluate the safety and efficacy of these targeted immunotherapies in clinical settings.

### 5.2. Personalized Treatment Strategies for MAP and SAP

Given the heterogeneous nature of AP, personalized treatment strategies that take into account disease severity (i.e., mild acute pancreatitis or severe acute pancreatitis) and individual immune profiles may be beneficial. For example, MAP patients who exhibit increased regulatory B cell populations may require different therapeutic interventions than SAP patients showing a depletion of these cells [92]. Personalized treatment strategies may involve the use of immune-modulating agents, immunosuppressive drugs, or immunostimulatory interventions based on the patient’s immune status and disease severity.

### 5.3. Combination Therapies and Immune Modulation

Combination therapies that involve multiple immunotherapeutic agents may enhance the efficacy of AP treatment. These approaches may target various aspects of the immune response, such as pro-inflammatory T cell subsets, regulatory T cells, B cells, and NK cells. For example, a combination of therapies that target both Th17 and Th22 cells may yield synergistic effects in modulating inflammation and tissue damage during AP. Additionally, the use of immune modulators, such as anti-inflammatory cytokines (e.g., IL-10, TGF-β) or agents that promote the differentiation and function of regulatory immune cells, may improve disease outcomes by reducing excessive inflammation and restoring immune homeostasis.

### 5.4. Addressing Knowledge Gaps and Future Research

Despite the recent advances in understanding the role of immune cells in AP, there remain significant knowledge gaps that warrant further investigation. These include elucidating the precise roles and mechanisms of various immune cell subsets in AP pathogenesis, understanding the complex interplay between innate and adaptive immune responses during AP, and identifying additional potential therapeutic targets. Longitudinal studies and comprehensive immune profiling of AP patients are needed to better understand the dynamics of immune cell populations throughout disease progression and recovery. Moreover, the development and validation of predictive biomarkers for disease severity and prognosis may enable more accurate patient stratification and personalized treatment approaches.

## 6. Conclusions

In conclusion, inflammation in acute pancreatitis begins with the abnormal activation of trypsinogen, leading to the injury of acinar cells and, in some patients, a systemic inflammatory response and even multiple organ dysfunction syndrome. In recent years, research has highlighted the complex interplay between various T cell subsets and B cells, which have unique roles in the pathogenesis of AP. Pancreatic acinar cell damage can occur directly through interactions between acquired immunity cells and pancreatic acinar cells, or indirectly by stimulating effector functions of innate immune cells by acquired immunity cells and promoting pancreatic damage. The balance between pro-inflammatory and anti-inflammatory immune responses is crucial in determining the outcome of AP, with the immunoregulatory functions of regulatory T and B cells playing significant roles in controlling excessive inflammation and maintaining immune tolerance. Understanding the precise roles of these immune cells in AP will help develop targeted therapeutic strategies to modulate the immune response and improve patient outcomes. Further research is needed to elucidate the roles of these immune cells in the development of AP and to evaluate the potential of targeting specific immune cell populations for treating this complex and severe inflammatory disorder.

## Figures and Tables

**Figure 1 cells-12-01495-f001:**
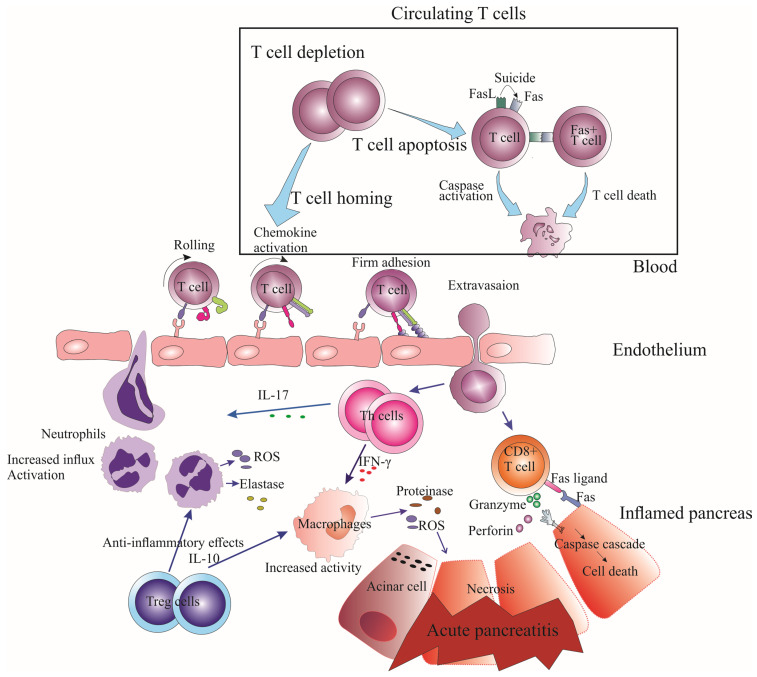
Mechanisms of T Lymphocyte Depletion and Pancreatic Damage in Acute Pancreatitis. This figure illustrates the depletion of circulating T lymphocytes in acute pancreatitis development due to the following two primary factors: increased apoptosis of T lymphocytes caused by heightened Fas/FasL signaling pathway and caspase activation, and enhanced T cell homing to inflamed tissues through rolling, chemokine-mediated activation, and firm adhesion. As a result, the inflamed pancreas experiences increased T cells that directly and indirectly damage the pancreatic parenchyma, with direct damage mediated by CD8^+^ T cells and FasL/Fas interactions, and indirect damage mediated by Th1 and Th17 cells that stimulate macrophage activation and neutrophil influx, respectively.

**Figure 2 cells-12-01495-f002:**
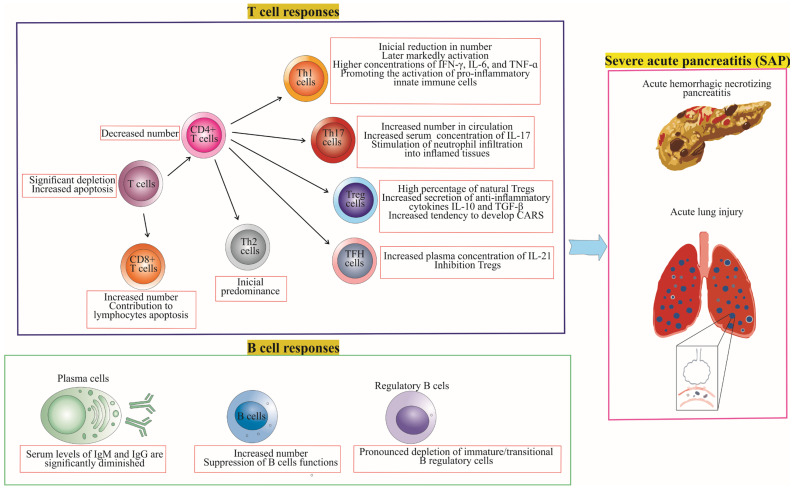
Adaptive Immune Response in Severe Acute Pancreatitis (SAP). The figure details the dynamic immune landscape during SAP, highlighting the decrease in T and helper T cells, increase in cytotoxic T and Th17 cells, and the shift from initial Th2 to later Th1 dominance. It underscores the role of Th1-derived cytokines (IFN-γ, IL-6, TNF-α) in activating pro-inflammatory innate cells, and the contribution of IL-17 in neutrophil recruitment. Furthermore, it illustrates the rise in IL-21 (a TFH cell marker) and the compensatory activation of Tregs, predisposing to CARS. B cell responses are marked by increased numbers but suppressed functions, alongside a decrease in regulatory B cells. These changes characterize SAP, facilitating local complications of acute pancreatitis such as hemorrhage and pancreatic necrosis, and systemic complications such as acute respiratory distress syndrome.

**Figure 3 cells-12-01495-f003:**
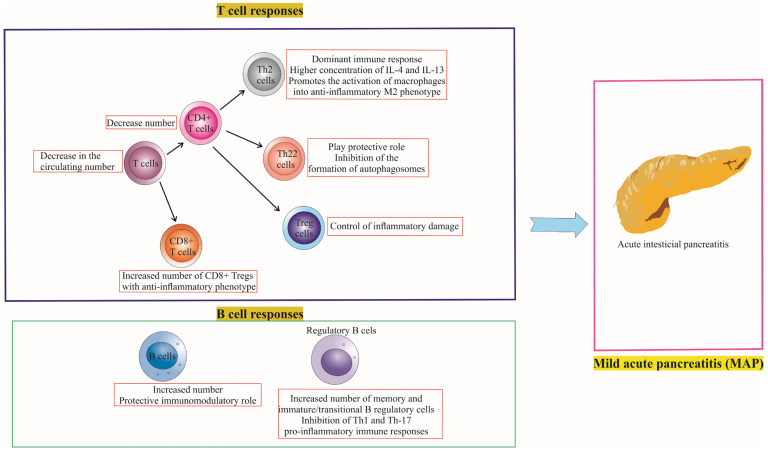
Immune Landscape in Mild Acute Pancreatitis (MAP). The figure succinctly captures the anti-inflammatory shift in MAP, marked by decreased T and Th cell counts, and an increase in Th2 cells with enhanced IL-4 and IL-13 secretion, favoring M2 macrophage polarization. It spotlights the protective role of Th22 cells and the increased activation of Tregs in mitigating inflammatory damage. Additionally, it notes the rise of cytotoxic T cells exhibiting anti-inflammatory phenotypes and B cells with a predominant immunomodulatory role, inhibiting Th1 and Th17 responses. These dynamics collectively characterize the mild AP course.

## Data Availability

The data presented in this study are derived from previously published research articles and publicly available sources. These data sources are cited throughout the manuscript, and interested readers can access them through the provided references. No new, original data were generated or analyzed for this study.

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
