# Peer review of "The Emerging Roles of the Adaptive Immune Response in Acute Pancreatitis"

_cells, 2023, doi:10.3390/cells12111495_

Round 1
Reviewer 1 Report
In this review, the authors highlighted the activation and the role of different populations of adaptive immune cells during acute pancreatitis and the impact of the pro- and anti-inflammatory cytokines on the progression of severe forms. This overview is interesting and provides many informations regarding the phenotypes of activated T cells in different stages of acute pancreatitis. This manuscript is well presented and deepens our knowledges in this area. 2 minor points needs to be corrected to improve the quality of the review.
- Line 57, “MAP” must be defined here (vs line 138)
- The figure 2 is too small to be correctly analyzed, specifically the text included in the figure is illegible. It is an important figure that illustrates all the information described in the different paragraphs of the review. Therefore it will be necessary to enlarge the figure to increase its clarity.
Author Response
Dear Reviewer 1,
Thank you for your positive feedback on our manuscript and for providing specific details on the minor points that need correction. We appreciate your recognition of our efforts in presenting a comprehensive overview of the role of adaptive immune cells and cytokines in acute pancreatitis. We have addressed your concerns as follows:
- Point: Line 57, “MAP” must be defined here (vs line 138)
Response: We agree with your suggestion and have now defined "MAP" (mild acute pancreatitis) at its first occurrence in the text (line 69)
- Point: The figure 2 is too small to be correctly analyzed, specifically the text included in the figure is illegible. It is an important figure that illustrates all the information described in the different paragraphs of the review. Therefore it will be necessary to enlarge the figure to increase its clarity.
Response: We appreciate your feedback on Figure 2. To enhance clarity and readability, we have decided to split it into two separate figures: Figure 2 focusing on severe acute pancreatitis (SAP) and Figure 3 on mild acute pancreatitis (MAP). We believe this separation will allow for better understanding and interpretation of the data.
We hope that these changes meet your expectations and further enhance the quality of our review. Once again, thank you for your valuable input.
Sincerely,
Bojan Stojanovic, MD, PhD
Assistant Professor of Surgery
Department of Surgery, Faculty of Medical Sciences, University of Kragujevac, Serbia
Reviewer 2 Report
The authors provide a in depth review of the role of lymphocytes in mild and severe acute pancreatitis with an emphasis on the impact of Th cells. Overall, the review is well organized but lack of descriptive clarity clouds the impact.
The authors need to add an initiral paragraph and/or diagram that clearly delineates and defines the types and stages of progression for acute pancreatitis and severe pancreatitis. This will orient readers and help as the authors compare and contrast the two conditions. The authors might want to exclude infectious pancreatitis in this review or make a separate section because it seems to have significant differences in immune response that complicate understanding of the pathophysiology of T cells in response to AP progression.
In multiple instances, it is difficult to know whether the authors are referring to the human condition or if they are referring to an experimental model and if so which experimental model as there are many ways to induce pancreatitis in mice and they do not always elicit the same response. Consistent clarification would improve the impact of this review.
Specific corrections to address:
Line 57: Define MAP
Lines 58-61: This sentence implicates AP but ends with severe forms of the disease. It is unclear if the authors are switching between progression of disease or types of disease.
T cells (lines 70-86). The author points to studies that implicate reduction of T cells early but then other studies that implicate reduction in severe AP, it is difficult to understand the overall implication the author is implying; is the first case MAP or are they talking about different types of T cells or the difference between circulating versus organ site.
Line 82-83. If the CD4/CD8 ratio is decreased then that implicates the CD8 cells might not subject to the Fas/FasL to the same extent as the CD4 cells? Please clarify which specific T cells are affected, is it one subset or both?
Figure 1 legend is confusing and needs clarifications. Please add “circulating” to legend (also lines 80-81). The figure overall is busy and a stepwise walkthrough would be helpful for the legend. For simplicity, I would remove the B cells from this figure as it is unclear why they are needed at this point in the review.
Conceptually, if T cells are depleted “outside” of the pancreas, then wouldn’t this also decrease the number of T cells getting to the pancreas. Why would homed T cells not be subject to the same apoptosis? Why would they be viable in once in the pancreas? How do they get to the pancreas if they are dying in the circulation?
Lines 93-99: Authors need to add reasoning/evidence for which mediators of adhesion are altered on T cells that directly affects a component of homing. Nothing is presented for pancreas.
Line 102 is confusing as it is mentioned the subset of CD4 cells increased after admission. Additionally, low numbers predicted serious disease. Is the increase in the first instance indirectly due to treatment of patients that have been admitted? or does the level go up and then down? Or never go up in the organ failure patients. Please clarify these statements and implications.
Line 110 please define abdominal compartment syndrome. How does this affect AP or why is ACS relevant?
Line 179-181 is not supported by evidence. Please include papers that have shown therapeutic suppression of TH17 in the setting of pancreatitits and mention how the TH17 was achieved for those studies.
Line 188-190 evidence seems to conflict with evidence in lines 186-188. Please clarify how/why some studies show increased IL22 in AP but yet somehow in the lungs it is reduced, this argues against a systemic alteration or potentially IL-22 coming from a non-Th22 source. The conclusion in 190-191 is therefore not reflective of both conditions.
Lines 191-193. Authors should more clearly delineate the connection between IL-22 or Th22 cells and the upregulation of Bcl-2. This is not clear without having to read the cited papers.
Lines 194-195. The connection between aryl hydrocarbon receptor is not clear, where is it coming from or how is it being altered? This statement seems random.
Section 2.1.5. What is the role of IL-21? How/why would it affect the progression of pancreatitis?
Lines 248-250 do not agree with one another. As stated Dbrowski shows decrease and you mention Pinhu also notes increased? Please clarify.
Figure 2. Consider reorganization by placing the outcome on the left side. I would also suggest putting the MAP on the top and the SAP on the bottom. In a progression model it usually starts with mild and moves toward severe. Hence the order of the figure should mirror that.
Future Directions: I would consider a section addressing this topic to present the major gaps you think should be addressed to propel the field forward. Maybe expanding upon specific immunotherapies? Or how one might treat AP different than SAP.
Author Response
Dear Reviewer 2,
We would like to express our gratitude for your thorough review and constructive feedback on our manuscript. Your insights have been invaluable in improving the clarity and organization of our work. In response to your comments, we have made several revisions and additions to the manuscript. Below, we have provided a point-to-point summary of our revisions in accordance with your suggestions:
Point: The authors need to add an initial paragraph and/or diagram that clearly delineates and defines the types and stages of progression for acute pancreatitis and severe pancreatitis.
Response: Thank you for your constructive suggestion. We have incorporated an initial paragraph to clearly outline the types and progression stages of acute pancreatitis and severe pancreatitis. This addition can be found on page 1, lines 32-41 of the revised manuscript.
Point: In multiple instances, it is difficult to know whether the authors are referring to the human condition or an experimental model.
Response: We appreciate your feedback and have made efforts to clarify this point throughout the manuscript. Whenever reference is made to either the human condition or an experimental model, we have now explicitly indicated this to ensure clarity for the readers.
Specific corrections:
Point: Line 57: Define MAP
Response: We apologize for any confusion. We have now explicitly defined MAP as mild acute pancreatitis in the text for clarity (page 2, line 69).
Point: Lines 58-61: This sentence implicates AP but ends with severe forms of the disease. It is unclear if the authors are switching between progression of disease or types of disease.
Response: We appreciate your observation and have taken steps to clarify this point in the manuscript (page 2, lines 61-77). We have rephrased the sentence to clearly distinguish between disease progression and types of disease.
Point: T cells (lines 70-86). The author points to studies that implicate reduction of T cells early but then other studies that implicate reduction in severe AP, it is difficult to understand the overall implication the author is implying; is the first case MAP or are they talking about different types of T cells or the difference between circulating versus organ site.
Response: We understand the difficulty and have revised this section to provide a clearer understanding of the implications of T cell reductions in relation to different disease stages (MAP or SAP) and T cell types (page 2, lines 70-77).
Point: Line 82-83. If the CD4/CD8 ratio is decreased then that implicates the CD8 cells might not subject to the Fas/FasL to the same extent as the CD4 cells? Please clarify which specific T cells are affected, is it one subset or both?
Response: We have clarified this in the revised manuscript by specifying which T cell subsets are affected by the decreased CD4/CD8 ratio (page 2, lines 95-98).
Figure 1 legend is confusing and needs clarifications. Please add “circulating” to legend (also lines 80-81). The figure overall is busy and a stepwise walkthrough would be helpful for the legend. For simplicity, I would remove the B cells from this figure as it is unclear why they are needed at this point in the review.
Response: We acknowledge your concerns and have revised Figure 1 and its legend accordingly. We have added the term "circulating" to the legend and provided a more detailed, stepwise explanation of the processes depicted. To simplify the figure, we have also removed the representation of B cells.
Point: Conceptually, if T cells are depleted “outside” of the pancreas, then wouldn’t this also decrease the number of T cells getting to the pancreas. Why would homed T cells not be subject to the same apoptosis? Why would they be viable in once in the pancreas? How do they get to the pancreas if they are dying in the circulation?
Response: Thank you for your insightful comment. While it is true that a decrease in circulating T cells could lead to a reduction of T cells reaching the pancreas in theory, the findings suggest a more complex scenario. Despite the general depletion of circulating T cells, it appears that T cells were still capable of reaching the inflamed pancreas. This may be due to enhanced chemotaxis and adhesion mechanisms that guide T cells to inflamed tissues, enabling a significant number of T cells to be recruited to the inflamed pancreas despite the decrease in the overall pool of circulating T cells. We will clarify this point in the revised manuscript to avoid any potential confusion, utilizing appropriate medical terminology (pages:3,4; lines: 121-148).
Lines 93-99: Authors need to add reasoning/evidence for which mediators of adhesion are altered on T cells that directly affects a component of homing. Nothing is presented for pancreas.
Response: Thank you for your suggestion. In the revised manuscript, we have included further details to substantiate the notion that specific mediators of adhesion, namely integrins and selectins, may influence T cell homing to the pancreas, playing a pivotal role in this process. Our intention in doing so is to provide a more thorough and comprehensive understanding of the mechanisms that guide T cell homing to the pancreas. We hope that these additions will strengthen the manuscript and enhance its impact (page 3, lines 130-140).
Point: Line 102 is confusing as it is mentioned the subset of CD4 cells increased after admission. Additionally, low numbers predicted serious disease. Is the increase in the first instance indirectly due to treatment of patients that have been admitted? or does the level go up and then down? Or never go up in the organ failure patients. Please clarify these statements and implications.
Response: Thank you for your insightful comment. The observed rise in circulating CD4+ T helper cells within five days of disease onset is a common occurrence in patients with acute pancreatitis (AP). However, this increase in CD4+ T cells does not necessarily correspond to improved outcomes in all cases. In fact, a reduced number of CD4+ T cells at admission can predict the onset of severe AP. Therefore, while less severe cases of AP may exhibit an increase in CD4+ T cell numbers within five days of onset, patients with severe AP and organ failure may not experience the same recovery in CD4+ T cell numbers. As for the potential impact of treatment on CD4+ T cell numbers, this remains an area that requires further research to be fully understood (page 4, lines: 149-61).
Line 110 please define abdominal compartment syndrome. How does this affect AP or why is ACS relevant?
Response: We have expanded the definition and explanation of abdominal compartment syndrome (ACS) in the text. We've clarified its relevance to AP and how it can affect the progression and severity of the condition (page 4, lines: 164-170).
Point: Line 179-181 is not supported by evidence. Please include papers that have shown therapeutic suppression of TH17 in the setting of pancreatitits and mention how the TH17 was achieved for those studies.
Response: We appreciate your comment and have updated the manuscript to include the relevant references. Specifically, we have cited work demonstrating the therapeutic suppression of the Th17 response in experimental models of acute pancreatitis. These studies have employed strategies such as targeting key cytokines like IL-17, IL-23, or IL-6, which are instrumental in driving Th17 responses. Blocking of these cytokines was followed by reduction of both pancreatic and systemic inflammation associated with acute pancreatitis. However, we note that these studies are largely preclinical, and further research is necessary to evaluate the efficacy and safety of these approaches in clinical settings. We have clarified these points in the revised manuscript (page 6, lines: 240-243).
Point: Line 188-190 evidence seems to conflict with evidence in lines 186-188. Please clarify how/why some studies show increased IL22 in AP but yet somehow in the lungs it is reduced, this argues against a systemic alteration or potentially IL-22 coming from a non-Th22 source. The conclusion in 190-191 is therefore not reflective of both conditions.
Response: Thank you for pointing out this seeming contradiction. The discrepancy in IL-22 levels in acute pancreatitis versus in the lungs can be attributed to the complex and context-dependent roles of Th22 cells and IL-22 in different tissues and disease stages. It is important to note that increased plasma levels of IL-22 during AP may reflect a systemic response to inflammation. However, the reduced expression of IL-22 in the lungs could be due to localized tissue-specific factors or the suppression of Th22 responses in certain conditions. Furthermore, it's critical to recognize that Th22 cells are not the only source of IL-22. Other immune cells, such as NK cells, innate lymphoid cells, and NKT cells, can also produce IL-22. These cells might potentially contribute to the increased plasma levels of IL-22 observed in AP patients. We have clarified these points in the revised manuscript to ensure a more nuanced understanding of the role of IL-22 and Th22 cells in AP and its complications (page 6, lines: 254-261).
Point: Lines 191-193. Authors should more clearly delineate the connection between IL-22 or Th22 cells and the upregulation of Bcl-2. This is not clear without having to read the cited papers.
Response: We appreciate your comment. We have further elucidated the relationship between IL-22 or Th22 cells and Bcl-2 upregulation, making it easier to understand without needing to delve into the cited papers (page 6, lines: 262-268).
Point: Lines 194-195. The connection between aryl hydrocarbon receptor is not clear, where is it coming from or how is it being altered? This statement seems random.
Response: We understand the confusion. We have clarified the connection between the aryl hydrocarbon receptor, its origin, and how it's being modified in the context of pancreatitis (page 6, lines: 268-273).
Point: Section 2.1.5. What is the role of IL-21? How/why would it affect the progression of pancreatitis?
Response: Thank you for pointing this out. We have expanded on the role of IL-21 and its potential impact on the progression of pancreatitis (page 7, lines: 285-291).
Point: Lines 248-250 do not agree with one another. As stated Dbrowski shows decrease and you mention Pinhu also notes increased? Please clarify.
Response: We appreciate your attention to detail. We have clarified the apparent discrepancy between Dbrowski's and Pinhu's findings, offering a potential explanation for the observed differences (page 8, lines: 336-341).
Point: Figure 2: Consider reorganization by placing the outcome on the left side.
Response: Thank you for your valuable suggestion. We have reorganized the content of the original Figure 2, resulting in two separate figures. The revised Figure 2 now specifically pertains to severe acute pancreatitis, while Figure 3 focuses on mild acute pancreatitis. We believe this reorganization provides a clearer and more focused visual representation of the different immune responses in these two conditions (pages 10,11)
Point: Future Directions: I would consider a section addressing this topic to present the major gaps you think should be addressed to propel the field forward.
Response: We value your suggestion. We've added a "Future Directions" section to highlight the major gaps in the field and suggest possible ways to address them, including specific immunotherapies and different treatment approaches for AP and SAP (pages 11,12, lines: 444-483).
We have addressed each of your specific corrections in detail in the revised manuscript, and we believe that these changes have significantly improved the quality and clarity of our work. We are confident that these revisions address your concerns and that the updated manuscript will be better suited for publication.
Once again, thank you for your time and invaluable input.
Sincerely,
Bojan Stojanovic, MD, PhD
Assistant Professor of Surgery
Department of Surgery, Faculty of Medical Sciences, University of Kragujevac, Serbia
Reviewer 3 Report
1. In the abstract of the article, the authors only mentioned T cells and did not mention B cells. B cells are also an important part of the article. I think should also summarize the role of B cells in acute pancreatitis in the abstract.
2. On line 57, the first use of the abbreviation MAP requires the full name to be written out.
3. In Figure 2, the authors compared the differences between different T cells and B cells in MAP and SAP. I think it is necessary to reflect the differences between the two more in the main text.
4. Text in Figure 2 is not legible, improve the quality of Figure 2.
5. A total of 79 references were annotated in the article, and what are the final 80-87
Author Response
Dear Reviewer 3,
Thank you for your valuable feedback on our manuscript. We appreciate your suggestions and have made revisions to address your concerns as follows:
Point: In the abstract of the article, the authors only mentioned T cells and did not mention B cells.
Response: We have revised the abstract to include a summary of the role of B cells in acute pancreatitis. We've emphasized their importance in conjunction with T cells to provide a more balanced view of the immune response in this condition.
Point: On line 57, the first use of the abbreviation MAP requires the full name to be written out.
Response: We agree with your suggestion and have spelled out "mild acute pancreatitis" at the first use of the abbreviation MAP on line 69.
Point: In Figure 2, the authors compared the differences between different T cells and B cells in MAP and SAP. I think it is necessary to reflect the differences between the two more in the main text.
Response: Thank you for your suggestion. We've expanded the discussion of the differences between T cells and B cells in MAP and SAP within the main body of the text, to provide a clearer comparison and further explain their roles in these conditions (page9, lines: 385-432)
Point: Text in Figure 2 is not legible, improve the quality of Figure 2.
Response: We appreciate your feedback. To improve legibility and focus, we have divided Figure 2 into two separate figures, each dedicated to one condition (MAP and SAP), thus making it easier to interpret.
Point: A total of 79 references were annotated in the article, and what are the final 80-87?
Response: We apologize for the confusion regarding the numbering of references. We have double-checked and corrected the reference list to ensure that all cited references are accurately numbered and annotated.
We believe these revisions have addressed your concerns and enhanced the quality of our manuscript. Once again, we thank you for your insightful comments and guidance throughout this process.
Sincerely,
Bojan Stojanovic, MD, PhD
Assistant Professor of Surgery
Department of Surgery, Faculty of Medical Sciences, University of Kragujevac, Serbia
Round 2
Reviewer 2 Report
Authors have addressed all the major concerns and presented a significantly improved review.
Minor revisions:
I would recommend expanding upon the description in Figure 2 and 3 legends for SAP and MAP to clearly explain what you are showing. It is not clear what is being implicated in the illustrations.
Author Response
Thank you for your insightful comment. Here are the revised figure legends, providing more clarity on what the figures represent:
Figure 2: Dynamics of Adaptive Immune Response in Severe Acute Pancreatitis (SAP). This figure provides a comprehensive overview of the immune cell fluctuations during the progression of SAP. It highlights the decrease in overall T cells, specifically helper T cells, contrasted by an increase in cytotoxic T cells, which play a crucial role in triggering lymphocyte apoptosis. The diagram depicts an initial dominance of Th2 cells, shifting to Th1 cells in later stages that secrete key pro-inflammatory cytokines (IFN-γ, IL-6, TNF-α), facilitating the activation of innate immunity cells. Furthermore, it shows an increase in Th17 cells and the secretion of IL-17, driving neutrophil migration into inflamed areas. IL-21, a signature cytokine of TFH cells, is also seen to increase, along with a compensatory rise in Tregs that may predispose to CARS. In terms of B cell responses, the figure illustrates an increase in B lymphocyte numbers but a suppression in their functional activity, as evidenced by decreased immunoglobulin secretion, along with a depletion of regulatory B cells. These changes underscore the pro-inflammatory bias in SAP, contributing to local and systemic complications.
Figure 3: Immune Modulation in Mild Acute Pancreatitis (MAP). This figure details the immune responses in MAP, marked by a decrease in overall T and Th cells, and an increase in Th2 cells secreting IL-4 and IL-13, fostering the development of anti-inflammatory M2 macrophage phenotypes. The protective role of activated Th22 cells is highlighted, inhibiting autophagosome formation, along with an increased activation of Tregs that assist in controlling inflammatory damage. The figure also shows a rise in cytotoxic T cells with anti-inflammatory phenotypes. B cell responses are characterized by increased numbers but with a predominant immunomodulatory role, demonstrated by their inhibitory action on Th1 and Th17 immune responses. These events collectively define the course of mild AP, featuring a predominantly anti-inflammatory response.